# A deep reinforcement based echo state network for network intrusion classification

Khorshed Alam[1], Mahbubul Haq Bhuiyan[1], Dewan Md. Farid[1,2]*

1 Department of Computer Science and Engineering, United International University, Dhaka, Bangladesh,
2 Department of Computer Science and Engineering, Southeast University, Dhaka, Bangladesh

☯ These authors contributed equally to this work.
* dewanfarid@seu.edu.bd

## Abstract

Network intrusion classification referred to the process of monitoring and analyzing network traffic to identify suspicious activities or attacks. In this work, author proposed a novel approach to classify network intrusion by utilizing deep reinforcement learning (DRL), integrating a reservoir computing approach Echo State Network (ESN). A DRL-based approach improved upon traditional deep learning by adapting dynamically to novel/unknown and evolving attack patterns. Unlike static models, DRL continuously learned optimal strategies through interaction with the environment, allowing for better detection of previously unseen threats in real-time. To address the class imbalance often encountered in network intrusion datasets, we evaluated the performance of several advanced data balancing techniques, including Borderline-SMOTE, SMOTE-ENN, ADYSN, and K-means SMOTE. The findings demonstrated that the K-means-based data balancing method outperformed other techniques, resulting in the most robust performance across various metrics. Author conducted multi-dataset validation on benchmark datasets like NF-BoT-IoT, NF-UNSW-NB15, NF-ToN-IoT, NF-ToN-IoT-v2, NF-CSE-CIC-IDS2018 and NF-UNSW-NB15-v3 to ensure robustness across different network flow data. For adaptive modeling testing, author excluded some attack types from training data and included them in testing data (e.g., DoS, Backdoor attacks were excluded from the training data but included in the testing data (see Table 3)). The proposed approach enhanced the accuracy and reliability of intrusion detection, making it a viable solution for securing modern network infrastructures. The source code of this work is available in this Github repository (https://github.com/codewithkhurshed/DRLZDNIDS).

## 1 Introduction

Network Intrusion Classification (NIC) plays a vital role in protecting computer networks by monitoring network traffic for signs of suspicious activity, such as malware,

**Data availability statement:** The data underlying the results presented in the study are available from the University of Queensland (https://staff.itee.uq.edu.au/marius/NIDS_datasets/).

**Funding:** The author(s) received no specific funding for this work.

**Competing interests:** he authors have declared that no competing interests exist.

unauthorized access, or intrusion attempts [1]. These systems analyze incoming and outgoing data in real-time, looking for anomalies or patterns that indicate potential threats. It works much like an early warning system to the network administrators against potential attacks, and appropriate responses can be done much before considerable damage is caused. Due to continuously evolving advancement of the cyber threats, the development of a perfect NID is of prime importance. Attackers continuously vary their techniques to get past traditional security measures like firewalls and antivirus programs. Ideally, the NID would do this not only for known attack patterns with high efficiency but also for novel threats never seen before with high accuracy. Besides, a faultless NID would reduce false negatives-when benign activities are flagged as attacks-which becomes very important in keeping unnecessary interventions at their minimum and ensuring no real threats slip by. Since it emerges more sophisticated and resilient, the better that NID can protect integrity, confidentiality, and availability of network resources, its development now forms one of the cornerstones in modern cybersecurity strategies.

Research in computer networking becomes vital because it forms the backbone of modern communication that runs everything from personal use of the internet to worldwide business operations. The complexity and scale of networks keep on increasing as newer technologies such as 5G, cloud computing, and IoT are stretching traditional infrastructure beyond its limits [2]. AI involvement in networking transforms how the networks will be managed, optimized, and kept secure. AI-powered solutions [3] enable intelligent traffic routing, predictive maintenance, and real-time intrusion detection, ensuring networks are resilient, efficient, and adaptive to dynamic conditions [4]. As network infrastructures evolve, ongoing research is essential to address challenges like scalability, security, and performance optimization, making it pivotal for both technological advancement and the seamless functioning of digital economies.

In the area of NIC, two types of methodologies have been identified in the previous study [5,6]. They are the DL-based methodology and the DRL-based methodology [7], with each having merit order based on design and application in use. This methodology from DL-based uses neural networks for processing data volumes that arise in network traffic. This will cause the system to detect and extract complex patterns and features. As the power of this technique rests in classification, it provides an easy pathway for DL-based NID to catch recurring or established attack patterns, considering their training would deal with high-quality substantial datasets of past attack knowledge. Thus, it is more appropriate and helpful in environments where high intrusion detection accuracy of known intrusions is required. While providing all-encompassing historical attack data, a DL-based NID will work just fine; these models become very good at finding recurring threats and well-documented attack signatures.

The DRL-based approach introduces an element of adaptability and continuous learning into intrusion detection. Unlike DL, which operates on static datasets, DRL involves an agent interacting with the network environment, receiving real-time feedback, and adjusting the detection strategy in pursuit of the same. This allows

DRL-based systems to adapt dynamically to variations in network conditions and to the emergence of new or unknown threats. The adaptability thus provided makes a DRL-based approach particularly suitable for environments where novel or evolving threats are common-for instance, in government agencies or financial institutions, where attackers frequently employ either custom or advanced attack methods. This ability of the DRL model to learn and readjust in real time gives it an edge in handling sophisticated and unpredictable intrusion attempts. DL-based NIC applies to organizations that require high accuracy on known threats by using historical data, whereas DRL-based NID applies in scenarios where organizations require systems that are flexible and adaptive in handling emerging threats differently. Both approaches boast unique strengths, and the choice depends on the specifics of each organization's requirements and the kind of threat it faces [7].

The DRL-based approach in NID is efficient in the detection of zero-day attacks due to its capability of continuous learning from the network environment [8]. In a typical DRL framework, there is an agent that perceives network traffic as a state and takes an action whether the observed traffic is a threat or not. In time, the agent receives feedback in terms of rewards or penalties depending on whether decisions taken are correct or not. This learning can enable the agent to dynamically refine the detection strategies and enhance the capability of detecting new patterns and anomalies in network flow that might indicate previously unknown attack techniques. While static models rely on preprogrammed rules and patterns, DRL systems can adapt in real time and may therefore detect sophisticated and changing cyber-attacks. Among the key advantages of the adaptive modeling features of DRL are that the system becomes enabled to continuously optimize intrusion detection capabilities when its exposure to new types of attacks increases. Most of the traditional systems require updates manually if some new type of attack is identified. In contrast, this is where DRL-based NID can excel: with its self-adapting behaviors and continuous learning from its environment, it can perform well in zero-day attacks or other evolving threats. Because of behaviors that differ from normal ones even though those behaviors were at first not recognized in its training set-the DRL model can identify the anomalies within the network data, not bound by static signatures or any predefined knowledge. This adaptability makes DRL-based NID very effective in maintaining security under constant changes within the threat landscape. The DRL-based systems continuously learn and adapt; therefore, they form a robust defense mechanism against known and unknown threats, bringing in proactive and resilient network security.

The motivation for using DRL in the network intrusion classification problem is because cyber threats are getting increasingly sophisticated and adaptable, which sometimes are not capable of being dealt with by a conventional IDS. Traditional methods, such as signature-based or rule-based systems, depend on predefined patterns and are usually static; thus, they easily become susceptible to novel evolving zero-day attacks. While DRL will provide an adaptive intrusion detection method wherein systems learn continuously through real-time interaction with the network environment. In addition, having the feedback mechanism follow naturally whereby the system gets rewarded or penalized for its actions will make the model iteratively refine its detection strategy toward emergent threats and anomalies. This adaptability ensures DRL-based systems will not be bound to predefined attack signatures but can generalize to detect previously unseen intrusion patterns. Besides, integrating DRL in network intrusion classification blends with modern complex network environments-like ones propelled by IoT, cloud computing, and 5G-technologies where the attack vectors change at high speed. The fact that DRL can balance high detection accuracy with minimal false positives, while at the same time adapting dynamically to ever-changing cyber threats, makes it a formidable choice for next-generation intrusion detection systems. This guarantees robust, proactive security in networking.

Author proposed a DRL based Network intrusion classification framework powered by Echo State Network architecture, which can detect novel attack variants of network flow data. The contribution of our work is stated below:

- Proposed a novel network intrusion classification framework integrating Deep Reinforcement Learning with Echo State Network (ESN), a reservoir computing method.

- Highlighted that the application of ESN in Network Intrusion Detection Systems is highly underexplored, motivating its integration for lightweight and efficient anomaly detection.

- Demonstrated that DRL enables dynamic adaptation to novel and evolving attack patterns, outperforming traditional static deep learning models (explained in section IV-B).

- Addressed class imbalance in network intrusion datasets by evaluating advanced data balancing techniques, including Borderline-SMOTE, SMOTE-ENN, ADYSN, and K-means SMOTE.

- Performed comprehensive multi-dataset validation using benchmark datasets such as NF-BoT-IoT, NF-UNSW-NB15, NF-ToN-IoT, NF-ToN-IoT-v2, NF-CSE-CIC-IDS2018, and NF-UNSW-NB15-v3.

- Designed an adaptive modeling test by excluding specific attack types (e.g., DoS, Backdoor) from the training set and including them only in the testing set to evaluate generalization capabilities (Shown in Table 3).

- Achieved improved detection accuracy and robustness, establishing the proposed DRL+ESN framework as a viable and efficient solution for network intrusion detection.

In Section I, the introduction emphasizes the growing need for advanced network intrusion detection systems and introduces the deep reinforcement learning approach as a solution to address the shortcomings of traditional methods. In Section II, the background provides foundational knowledge on deep reinforcement learning set up and data in the context of network security. In Section III, the related work reviews existing literature on intrusion detection systems, identifying gaps in prior research and highlighting the need for the proposed DRL-based model. In Section IV, the methodology outlines the design of the DRL-based NIDS, detailing the architecture, datasets, and techniques used for feature selection and model training. In Section V, result analysis presents experimental findings, comparing the proposed model's performance with existing methods, showcasing improvements in detecting known and novel attacks. In Section VI, the testing section evaluates the model's adaptive capabilities using the NF-BoT-IoT dataset, demonstrating its effectiveness in detecting previously unseen threats. In Section VII, the discussion focuses on the real-time adaptability of the DRL-based system and its broader implications for network security. In Section VIII, limitations are acknowledged, such as the complexity of implementation and the computational demands of training deep reinforcement learning models. In Section IX, the conclusion summarizes the contributions, highlighting the effectiveness of the DRL-based NIDS and suggesting future research directions for further improvement.

## 2 Background

### 2.1 Deep reinforcement learning (DRL)

Deep Reinforcement Learning comprises a sophisticated approach, combining two of the most powerful fields in artificial intelligence: reinforcement learning and deep learning. Reinforcement learning is a paradigm of machine learning wherein an agent learns to make decisions through interaction with an environment. The agent seeks to maximize some notion of cumulative reward through trial and error, whereby its behavior is modified based on feedback received from the environment. This includes taking actions in different states of the environment and getting the outcomes from these, which become helpful for later decision making. With time, the agent develops its own strategy, called policy, to act more successfully with regard to its goals [9]. Deep learning, on the other hand, is a subset of machine learning that makes use of multi-layer neural networks, hence giving the system capabilities to model and process highly complex data structures such as images, text, and sequences. Employing deep learning, DRL allows agents to handle environments where observations are high-dimensional input data, including long sequences in time-series data. In DRL, the neural networks are used to approximate value functions or policies that enable the agent to generalize over very large state and action spaces, and as such, enable the solving of hard problems that are otherwise intractable by traditional RL methods. Deep

reinforcement learning integrates the decision-making framework of reinforcement learning with the rich representation capabilities of deep learning and, therefore, is one of the main techniques used for solving complex tasks. In the process of DRL, the agent receives feedback as rewards or penalties based on the actions that it performs, thus enabling it to improve its performance over time. Before diving into DRL, it is crucial to understand the basic components of Reinforcement Learning:

- Agent: The learner or decision-maker that interacts with an environment.

- Environment: The external system the agent interacts with, which provides feedback.

- State: The current situation of the agent in the environment.

- Action: The decisions the agent can make.

- Reward: A scalar feedback signal received after taking an action in a particular state. The goal of the agent is to maximize the cumulative reward over time (also known as the return).

- Policy: A strategy that the agent follows to choose actions based on its current state.

- Value Function: A function that estimates how good a particular state or action is in terms of long-term rewards.

- Q-value (Action-Value): The expected return of taking an action in a given state and following a policy thereafter.

The agent interacts with the environment in discrete time steps: at each step, it observes the state, selects an action according to its policy, receives a reward, and transitions to a new state. This continuous feedback loop helps the agent improve its policy over time.

**2.1.1 Network intrusion classification.** DRL can substantially enhance Network Intrusion Classification by enabling adaptive and autonomous learning in the presence of evolving cyber threats. Unlike traditional NIC approaches that rely on static signatures or predefined rules and are therefore limited to known attack patterns, DRL-based systems learn directly from network traffic through continuous interaction with the environment. By leveraging neural networks to model high-dimensional and complex traffic data, DRL agents can extract rich representations of normal and malicious behavior, allowing them to identify subtle anomalies and previously unseen attack vectors, including zero-day attacks. This dynamic and self-improving learning process makes DRL-enhanced NIC systems more proactive and resilient, as they continuously refine their detection policies in response to emerging and increasingly sophisticated threats.

**2.1.2 Adaptive learning by DRL.** DRL enables an efficient and adaptive intrusion response strategy by overcoming the rigidity of traditional security mechanisms that rely on predefined and often inflexible reaction policies. Instead of applying static responses, a DRL agent dynamically selects the most appropriate mitigation action based on the detected threat's characteristics, severity, and the current network state. Leveraging prior experience learned through interaction with diverse attack scenarios, the agent can enact lightweight responses—such as selectively blocking suspicious IP traffic—in minor cases, while escalating to more disruptive but necessary actions in critical situations, including traffic rerouting to maintain service continuity during active attacks. This adaptive decision-making framework allows DRL-based systems to respond proportionally and intelligently to a wide spectrum of intrusion events.

In addition, the agent can take rate-limiting measures by reducing the bandwidth or limiting the number of requests coming from suspicious sources. This mechanism prevents Distributed Denial of Service (DDoS) and other threats that occur in high volume from causing much damage-throttling such malicious traffic while allowing legitimate users access to the network resources. In addition to such an immediate response, DRL can contribute even in the higher-order task of resource deployment optimization across a network during an attack. For instance, while an attack is in progress, the network must be efficient and, if needed, declare its defense. A DRL agent may be designed to understand which parts of the network are most critical during the attack, and, accordingly, focus the allocation of additional security. For example,

the agent will be able to identify which server or device is at a higher risk to be isolated from the other network so it will not propagate malware and leak information.

Should the attack be highly focused, the DRL system can go all the way to determine which network parts should be shut down or temporarily isolated in order to reduce the scope for widespread damage. It learns to take such decisions through balancing the trade-offs between preserving operational functionality and the prevention of further attack escalation. The power of adaptability ensures an effective response and one optimally designed to minimize its impact on business critical operations. DRL can be used to continually modify the response strategy over time. With experience gained by the agent, responding against other types of intrusion, the agent further refines its decision process and thus becomes more efficient in choosing responses that minimize further damage, make effective resource utilization, and preserve the functionality of key network components. This makes DRL a potent method for developing smart, independent cybersecurity systems capable of adapting to an ever-changing threat landscape.

### 2.1.3 DRL's strength in handling unseen attacks.

**Generalization through Policy Learning:** DRL models do not merely memorize patterns seen during training; they learn a policy, a set of actions or decisions, based on the broader characteristics of network traffic behavior. These policies are trained to maximize rewards (successful detection of intrusions) by adapting to various network conditions, making the model capable of generalizing to unseen attack types. Even though DDoS and Injection attacks were not part of the training data, the model can recognize patterns similar to previously encountered attacks (Backdoor, denial-of-service (DoS)), utitilizing its policy to identify these new threats as anomalies or malicious.

**Exploration and Exploitation:** In DRL, the exploration-exploitation trade-off allows the model to explore different actions and outcomes in its training phase. This enables the model to adapt to new attack vectors in the test set, as it can identify similarities in network flow anomalies, even if the precise attack type (e.g., DDoS or Injection) was not in the training data. Through exploration, the DRL model learns features that are common across various attack types and exploits this knowledge to detect unseen threats.

**Feature Extraction:** Deep learning models, including DRL, are highly effective at extracting important features from raw network flow data. DRL can learn abstract representations of network behaviors, enabling it to differentiate between benign and malicious activities even when new types of attacks arise. For instance, DDoS and Injection attacks might have different signatures than Backdoor or DoS, but they may share certain characteristics such as abnormal traffic spikes, unusual request patterns, or irregular data injections, all of which the model can pick up during testing.

**Adaptation via Continuous Learning:** DRL models are inherently capable of continuous learning through interaction with their environment. Although the training phase is fixed, the DRL model's architecture allows it to adapt to new situations by leveraging its understanding of previously encountered malicious behavior. This adaptive quality is crucial for dynamic environments like networks, where new types of attacks are constantly emerging.

## 2.2 Analyzing network traffic patterns

We have used NF-Bot-Iot as primary dataset. Let us understand the data and features before diving into intrusion detection. NF-Bot-IoT [10] dataset represents network traffic information, capturing various aspects of packets exchanged between devices in a network. The dataset captures both normal (benign) traffic and malicious activity (thefts). By analyzing these patterns, it becomes possible to identify anomalies and potential threats. The presence of various transport layer protocols (TCP, UDP) indicates the diversity of applications in use. Notably, common ports such as 80 (HTTP) and 53 (DNS) are frequently encountered, which reflects typical web and network behavior. Fields such as IN_BYTES and OUT_BYTES can be used to analyze the amount of data transmitted during each session. This is critical for assessing the impact of specific traffic flows on network performance. TCP_FLAGS and FLOW_DURATION_MILLISECONDS provide insights into the connection dynamics, helping to distinguish between normal handshakes and potentially malicious patterns like SYN floods.

### 2.2.1 Detailed feature descriptions.

- IPV4_SRC_ADDR: This column represents the IPv4 address of the source device that initiates the data flow. IPv4 addresses are numerical labels assigned to each device connected to a computer network that uses the Internet Protocol for communication.

- L4_SRC_PORT: The source port number used by the application layer on the originating device. It identifies the specific service or process generating the traffic. Each service on a device is assigned a unique port number. For instance, web servers typically use port 80 for HTTP. Monitoring source ports can help in identifying unusual traffic patterns, such as services that are not typically accessed or unexpected connections.

- IPV4_DST_ADDR: This is the destination IPv4 address, indicating the target device that is receiving the data. Similar to the source address, the destination IP is crucial for mapping traffic flow within the network. It helps in identifying which devices are being accessed or attacked and can assist in pinpointing compromised devices.

- L4_DST_PORT: The destination port number at the transport layer, specifying the service or application on the target device that is intended to receive the data. Analyzing destination ports helps in understanding what services are being accessed. For example, if multiple flows target an unusual port, this might indicate probing for vulnerabilities or an ongoing attack.

- PROTOCOL: This represents the application layer protocol being utilized for the traffic flow, also denoted numerically. The application layer protocol provides insights into the type of service being accessed (e.g., HTTP, FTP). This helps in categorizing the traffic and understanding user behavior, as well as identifying any unusual or unauthorized applications in use.

- IN_BYTES: This column records the total number of bytes received by the source device during the data flow. Monitoring inbound traffic volume is critical for assessing network load and can highlight potential issues such as denial-of-service attacks, where a device may receive an abnormally high amount of data.

- OUT_BYTES: The total number of bytes sent out from the source device during the data flow. Similar to IN_BYTES, analyzing outbound traffic helps to monitor data ex-filtration attempts, where sensitive data may be sent outside the organization.

- IN_PKTS: The count of packets received by the source device during the flow. Packet counts can provide insight into the nature of the traffic. For instance, a high number of packets with low byte size might indicate a scanning or probing activity.

- OUT_PKTS: The total number of packets sent out from the source device. This metric is crucial for understanding how much data is being communicated and helps in identifying unusual patterns that could indicate malicious behavior.

- TCP_FLAGS: This field represents various flags set in the TCP header, such as SYN, ACK, FIN, and RST. These flags control the state of the TCP connection. Analyzing TCP flags is important for diagnosing connection states. For example, a high number of SYN packets without corresponding ACKs could suggest a SYN flood attack, indicating an attempt to overwhelm a service.

- FLOW_DURATION_MILLISECONDS: The duration of the data flow in milliseconds, indicating how long the flow lasted from start to finish. Flow duration helps in assessing the nature of the communication. Very short flows might indicate scanning, while longer flows could represent normal sessions. This can also help in identifying potential issues like timeouts or abrupt connection drops.

- Label: This categorical label provides an indication of whether the flow is considered benign or malicious (e.g., 'Theft').

- Attack: A binary indicator (0 or 1) that signifies whether the flow is associated with malicious activity (1) or is benign (0).

# 3 Related work

In this section, the related work reviews existing literature on intrusion detection systems, identifying gaps in prior research and highlighting the need for the proposed DRL-based model. Author selected papers from different network environment based NIC work as this study is focused on developing network intrusion classifier which can adapt on variety of network flow data by the help of DRL framework.

Recent progress in AI, more so DRL, has proved to be quite capable of improving the efficiency of NID. Benaddi et al. [11] have demonstrated the capability of RL in finding abnormal network access and attacks, a crucial task toward robust network security. Additionally, Lansky et al. [12] and Wang et al. [13] have underscored the efficacy of DRL in addressing these complex security issues. Current NID face significant challenges in detecting sophisticated and evolving cyber threats. Traditional intrusion detection methods often struggle with the complexity and volume of network data, leading to issues such as high false negative rates and the inability to detect novel attack patterns. Machine learning-based approaches, including network packet classification, have improved the performance and security of networks, but there remain gaps in their ability to handle large-scale, complex data and sophisticated attacks [14]. DRL based approach showed promising result, He et al. [15] proposed a transferable and adaptable network intrusion detection system (TA-NID) based on deep reinforcement learning. Their system improves robustness and adaptability by using small-scale datasets to generate diverse interactions, prioritizing outliers without requiring full dataset classification, and ensuring model transferability across different datasets. Experimental results showed high accuracy and effective prioritization of outliers. Dong et al. [16] proposed network abnormal traffic detection model Based on Semi-Supervised Deep Reinforcement Learning. The abnormal traffic detection can be further optimized with the utilization of the SSDDQN-based optimization approach. The core of this approach is how DDQN is applied to enhance the efficiency and accuracy of learning. An autoencoder was utilized to conduct feature reconstruction, thereby allowing the current network to conduct dimensionality reduction with the preservation of the key features of traffic. This is crucial for the processing of high-dimensional data where only the most important features remain for further analysis. Following feature reconstruction, accuracy in detection is enhanced through the classification by a deep neural network classifier. This classifier is enhanced as a result of the capability of the autoencoder on dimensionality reduction, which makes it concentrate more on the most important aspects of the traffic data. Besides, unsupervised learning is also included in the target network by incorporating K-Means clustering in order to group similar data into meaningful clusters. The DNN uses these clusters for making the prediction, thereby combining strengths from both unsupervised and supervised learning to further enhance the detection capability of abnormal traffic.

Kim and Pak [17] suggested approach involves generating a fresh training dataset for Generative Adversarial Network by leveraging misclassified data sourced from a bespoke training dataset via an LSTM-DNN model trained on the original dataset. There is hardware dependency and accuracy need to be further improved, compared to the conventional approaches. Machine learning based for network intrusion detection proposed by Santos et al. [18], achieving high accuracy but struggling with evolving traffic patterns over time. Commonly, these models require frequent updates, posing practical challenges. A novel model, using reinforcement learning for long-term reliability and classification accuracy, addresses these issues. This approach combines transfer learning and a sliding window mechanism to minimize computational demands and manual interventions. Experiments with an 8TB, four-year dataset reveal that traditional models less effective with dynamic traffic, whereas the proposed RL model maintains similar accuracy without frequent updates. Periodic updates further reduce false negatives, using significantly fewer resources and only seven days of training data. Ma and Shi [19] introduced a novel framework for anomaly detection in IDS, combining RL with class-imbalance techniques to improve detection accuracy. The methodology included an adapted Synthetic Minority Over-sampling Technique (SMOTE) to address dataset imbalance, enhancing the RL agent's performance. Experiments using the NSL-KDD dataset demonstrate the model's efficacy.

Comparative evaluations highlight that the proposed AESMOTE model outperforms AE-RL, achieving an accuracy above 0.82 and an F1 score above 0.824. However, limitations include reliance on the NSL-KDD dataset, which may not represent all real-world scenarios. Multi-dataset validation can help to prove proposed solution's robustness in terms of real-world scenarios.

MalBoT-DRL, a novel deep reinforcement learning-based botnet detector proposed by Al-Fawa'reh et al. [20], it allows dynamic adaptation to changing malware patterns. Validated on MedBIoT and N-BaIoT datasets, MalBoT-DRL achieved impressive detection rates of 99.80% and 99.40% in early and late detection phases, respectively. However, limitations include challenges in detecting stealthy malware like Bashlite, vulnerability to adversarial attacks, and increased latency with larger datasets. Despite these, the model maintained consistent training efficiency. Enhancing exploration-exploitation policies and employing network distillation can mitigate these challenges, ensuring robustness and reduced computational demands for real-world IoT environments. Picard and Pierre [21] proposed RLAuth, a risk-based authentication system using DRL to classify authentication contexts as anomalies or regularities. The methodology includes an Anomaly Detector and a Risk Engine to compute authentication risk based on context, previous authentication data, and application sensitivity. An Authentication Manager dynamically selects appropriate authentication methods. However, RLAuth is vulnerable in familiar contexts, such as attacks by coworkers or family members, and arbitrarily chosen confidence levels and application sensitivities could be optimized using mathematical or machine learning models. Additionally, storing context information on device memory leads to high consumption over time, suggesting a need for efficient modeling techniques. Despite these limitations, RLAuth enhances mobile application security by balancing usability and privacy, dynamically adapting to new environments. The Anomaly Detector achieved a G-Mean of 92.62% on the testing set, demonstrating the effectiveness of DRL in user authentication.

The study by [22,23] proposed a reinforcement learning-based collaborative DDoS detection method, using edge computing for early detection and efficient resource utilization. The methodology involved deploying lightweight unsupervised classifiers in IoT edge gateways to analyze network traffic features in real time. The soft actor-critic (SAC) reinforcement learning model dynamically adjusts classifier parameters, ensuring robust detection across various IoT devices. A collaborative aggregation module enhances detection by sharing states and experiences. Experiments on public datasets and a real-world dataset show excellent detection performance, accurately identifying stealthy IoT-based DDoS attacks. Limitations include potential adaptation delays in highly dynamic environments. Intrusion detectors are increasingly vulnerable to adversarial attacks, with current countermeasures often lower performance and being algorithm specific. Apruzzese et al [24] proposed a novel framework using DRL to protect botnet detectors from adversarial attacks. The methodology involves generating realistic adversarial samples that evade detection and using these samples to create an augmented training set, producing more resilient detectors. Extensive experiments on several machine learning algorithms and publicly available datasets validate the effectiveness of the proposed method. The results show significant improvements over state-of-the-art methods while being robust to unforeseen attacks and preserving performance in situations without adversarial manipulations. Key limitations include potential computational complexities as well as the requirement for heterogeneous training datasets.

The Industrial Internet of Things presents several cybersecurity risks, calling for advanced intrusion detection systems. Yu et al. [25] proposed a DC-IDS solution for IIoT using DRL, modeling the open-set recognition problem as a discrete-time Markov decision process. A DQN with a conditional variational autoencoder handles the known traffic classification and unknown attack recognition. Known traffic classification is handled by DQN while unknown attacks are recognized by reconstruction error. Investigations utilizing the TON-IoT dataset illustrate the efficacy of the DC-IDS model, which attains enhanced performance in identifying unfamiliar attacks while maintaining model stability relative to previous approaches. Potential constraints may encompass scalability issues associated with extensive IIoT implementations as well as the necessity for considerations regarding real-time application. Progressions in IoT and cloud computing introduce difficulties for intrusion detection within enterprise networks.

Vadigi et al. [26] proposed a Federated DRL-based IDS, deploying multiple agents with DQN logic across the network. Their system preserved data privacy by keeping data at each agent node local while utilizing an attention-weighted model aggregation process to share learning. An attention mechanism dynamically determines each agent's contribution to model aggregation. Tested on the ISOT-CID and NSL-KDD datasets, their system demonstrates robust performance with high accuracy, precision, and a low false-positive rate. Limitations include potential computational complexity and the need for effective handling of diverse network environments.

The work by Chavali et el. [27] introduced TD3-AP and SAC-AP, two DRL-based off-policy actor-critic methods for alert prioritization, addressing the instability and limited exploration of existing DDPG-based methods. Using twin delayed deep deterministic policy gradient (TD3) and soft actor-critic (SAC), the interaction between adversary and defender is modeled as a zero-sum game with a double oracle framework for mixed strategy Nash equilibrium. Extensive experiments on MQTT-IoT-IDS2020, DARPA 2000 LLDOS 1.0, and CSE-CIC-IDS2018 datasets show TD3-AP and SAC-AP reduce defender's loss by 50% and 14.28%, respectively, outperforming DDPG and traditional methods. The results' interpretability is enhanced using SHapley Additive explanations (SHAP). Limitations include sensitivity to hyperparameters and the need for significant training time and computational resources. Rookard and Khojandi [28] introduced RRIoT, utilizing a Deep Deterministic Policy Gradient (DDPG) reinforcement learning algorithm combined with an LSTM layer in an adversarial setting to enhance attack detection and identification. Compared to existing ML and RL algorithms like DQN, DDQN, and DDPG, RRIoT demonstrates superior performance. Evaluations using three IoT telemetry datasets show that RRIoT outperforms state-of-the-art ML algorithms and matches or exceeds novel RL algorithms. Additionally, Shapley Additive Global Importance (SAGE) is used to identify key contributing features, with feature importance confirmed through an ablation study. Limitations may include computational complexity and resource requirements for training the model. Lastly, the use of DRL in NID is particularly advantageous due to its ability to process and learn from vast amounts of data, identify complex patterns, and adapt to new and unseen threats. This makes DRL-based approaches well-suited to the demands of modern network security, where traditional methods often fail. As cyber threats continue to evolve, the integration of DRL into NID represents a significant advancement, offering enhanced detection capabilities and improved resilience against sophisticated attacks. Notably, intrusion detection using DRL techniques is still in development but shows great promise [29,30]. Researchers often aim for higher classification accuracy by modeling the intrusion detection domain as an DRL-based environment. Therefore, it is evident that DRL-based approaches are not only fitting but also necessary for the future of network intrusion detection.

A growing body of research highlights the significant advancements in applying GANs to intrusion detection systems. The work in [31] introduced a projection-based adversarial attack generation for IDS using a traffic space GAN to model the distribution of malicious and benign traffic. Similarly, [32] applied GANs to generate synthetic data, followed by classification using deep neural networks, convolutional neural networks (CNN), and long short-term memory, demonstrating that the GAN+DNN combination provided superior performance on the UNSW-NB15 dataset. Studies like [33] have addressed challenges in GAN training, such as instability and mode collapse, by proposing ensemble-based multi-layer GANs (EMP-GANs) that enhance training stability and synthetic data diversity. Meanwhile, traditional AI methods like decision trees [34–36] and support vector machines [37] have been widely explored in early network intrusion detection efforts. A comparative study in [38] evaluated Naïve Bayes, SVM, and K-Nearest Neighbors on NSL-KDD and UNSW-NB15 datasets, revealing varied results in intrusion detection accuracy across algorithms.

Deep learning approaches continue to gain traction in this domain, as demonstrated in [39], where DNNs showed their potential in efficiently classifying network threats. Zhong et al. [40] introduced a system integrating big data and tree structures with deep learning, yielding better performance compared to traditional methods. authors in [41] developed a method using GAN to produce adversarial attack data and proposed a robust IDS model (RAIDS) to counter these attacks, illustrating the evolving nature of threats and the need for adaptive defense mechanisms. In another study, [42] proposed a Convolutional Deep Belief Networks (CDBN) model for detecting intrusions in wireless networks,

which, despite its accuracy, is resource-intensive due to the complexity of its architecture. Recent hybrid models, like the CNN-BiLSTM architecture discussed in [43,44], further illustrate the potential of combining different deep learning techniques to handle both binary and multiclass intrusion detection, achieving an average accuracy of 84.42% on the NF-UNSW-NB15 dataset. Haghighat et al. [45] proposed an intrusion detection system based on deep learning and voting mechanisms that aggregate the best models for more accurate and robust results, effectively leveraging ensemble learning to counter diverse intrusion tactics.

Anomaly-based intrusion detection was contributed by [46,47], which proposed the Locality-Sensitive Hashing-based Isolation Forest model. With adaptive learning techniques, this model evolves continuously against evolving network conditions and is found to be very effective in highly variable environments. The system also applies window sliding and dynamic model updates of the features that may help in sustaining high accuracy over time. It is flexible enough to detect new and known threats efficiently while the nature of network traffic is continuously changing. This approach was tested with the widely used KDDCUP99 dataset and showed significant improvements in detecting anomaly activities compared to traditional approaches. It highlights the potential that adaptive learning bears with respect to the improvement of evolved cyber-attack detection. The proposed HC-DTTWSVM methodology for network intrusion detection by authors from [48] requires data collection and data preprocessing of network traffic from benchmark datasets like the NSL-KDD and UNSW-NB15, ensuring the quality of data by cleaning and normalization with feature extraction. Further, a bottom-up hierarchical clustering algorithm arranges this into a decision tree that optimally separates different intrusion categories. The TWSVMs are then integrated into this tree for fast classification of network traffic. The model will be trained using cross-validation techniques and tested against various state-of-the-art techniques on relevant performance metrics accuracy and F1-score. However, the approach suffers from some setbacks, including dependence on the quality of data, scalability issues when the dataset gets bigger, challenges in feature selection, and reduced generalization to new types of attacks not present in the training data. Moreover, to achieve the best result, hyperparameter tuning is required, and embedding TWSVM complicates the interpretability of the model, which may challenge stakeholders who require transparency in decision-making processes.

Similarly, the work in [49] demonstrates the application of Generative Adversarial Networks to improve the performance of intrusion detection systems in Internet of Things networks. Given the rapid expansion of IoT devices and the increasing diversity of potential security threats they face, traditional intrusion detection models often struggle with imbalanced datasets, where certain types of attacks are underrepresented. To address this, the authors employed GANs to generate synthetic data for these underrepresented attack classes, effectively balancing the dataset. This augmentation process resulted in a significant performance boost, improving the accuracy of the IDS from 84% to 91% when tested on the UNSW-NB15 dataset. The use of GANs in this context not only enhances the model's ability to generalize across different types of intrusions but also underscores the broader trend toward leveraging synthetic data generation in network security.

These advances emphasize the growing importance of deep learning techniques, particularly GANs, in addressing the dynamic and ever-evolving landscape of network threats. The ability to generate realistic synthetic data and adapt to changes in network traffic patterns represents a crucial advantage in modern IDS development. As cyber-attacks become more sophisticated, techniques like those presented in [46] and [49,50] provide essential tools for detecting anomalies and securing networks in a wide variety of contexts, from traditional networks to the more complex and heterogeneous environments of IoT systems.

The Table 1 presents a comparative analysis of various studies focused on Network Intrusion classification based on key features such as class imbalance handling, adaptive modeling, unseen threat detection, and the ability to manage evolving attacks. Each row corresponds to a different research paper, with the columns indicating whether the respective study effectively addresses these critical challenges. For instance, the works by Benaddi et al. [11], Lansky et al. [12], and Wang et al. [13] do not utilize any of the specified capabilities, suggesting that these approaches may be less effective in practical applications where cyber threats are continually evolving. He et al. [15] present an approach that could

**Table 1. Comparison of network intrusion classification based on key features.**

| Paper Ref. | Class Imbalance Handling | Adaptive Modeling | Unseen Threat Detection | Handling Evolving Attacks |
|---|---|---|---|---|
| [11] | No | No | No | No |
| [12] | No | No | No | No |
| [13] | No | No | No | No |
| [15] | No | Yes | Yes | Yes |
| [16] | Yes | Yes | Yes | Yes |
| [17] | No | No | No | Yes |
| [18] | No | No | No | Yes |
| [19] | Yes | Yes | Yes | Yes |
| [20] | No | Yes | No | Yes |
| [21] | No | Yes | No | Yes |
| [22] | No | Yes | Yes | Yes |
| [24] | No | Yes | Yes | Yes |
| [25] | No | Yes | Yes | Yes |
| [26] | No | Yes | No | Yes |
| [27] | No | Yes | No | Yes |
| **Proposed System** | **Yes** | **Yes** | **Yes** | **Yes** |

handle both the above challenges and include some form of adaptive modeling and unseen threat detection. It also fails to address the issues of class imbalance and management of evolving attacks. Similarly, the approach proposed in works like those by Kim and Pak [17], Vadigi et al. [26] suffers from either class imbalance or unseen threat detection challenges. It is observed that the work of Chavali et al. [27], while it may be efficient in adaptive modeling, does not have the capability of unseen threat detection, which again shows the further gaps in the existing methodologies. Of particular note, the final row represents the proposed system, which effectively manages all four features in an indication of comprehensiveness against the shortcomings identified above. It's indicative that the proposed system may result in key advances in intrusion detection, considering that solutions are desperately needed to be adaptable under shifting network conditions, handle class imbalances, and find unseen and evolving threats.

## 4 Methodology

Our methodology is based on the development of DRL based Network Intrusion detection. Our methodology consists of 3 steps. We have discussed each steps in detailed in following sub sections.

### 4.1 Data collection

We have collected dataset from open source NIDS datasets from University of Queensland [10] named NF-BoT-IoT, NF-UNSW-NB15, NF-ToN-IoT, NF-ToN-IoT-v2, NF-CSE-CIC-IDS2018, and NF-UNSW-NB15-v3. Each dataset consists of benign and attack data. Benign data are categorized as 0 and all types of attacks are categorized as 1. Our primary dataset is NF-BoT-IoT and rest of the datasets are used for multi-dataset validation task. NF-BoT-IoT consists of a total number of data flows of 600,100 out of which 586,241 (97.69%) are attack samples (Reconnaissance, DDoS, DoS, Theft) and 13,859 (2.31%) are benign. In NF-ToN-IoT, contains a total of 1,379,274 data flows, with 1,108,995 (80.4%) categorized as attack instances (Backdoor, DoS, DDoS, Injection, MITM, Password, Ransomware, Scanning, XSS) and 270,279 (19.6%) classified as benign. NF-UNSW-NB15 comprises 1,623,118 data flows in total, with 72,406 (4.46%) identified as attack samples (Fuzzers, Analysis, Backdoor, DoS, Exploits, Generic, Reconnaissance, Shellcode, Worms) while the remaining 1,550,712 (95.54%) are classified as benign. The NF-ToN-IoT-v2, NF-CSE-CIC-IDS2018, and

NF-UNSW-NB15-v3 datasets are NetFlow-based benchmarks designed to facilitate the development and evaluation of network intrusion detection systems (NIDS). NF-ToN-IoT-v2 comprises 16,940,496 flows, with 63.99% labeled as attack samples, reflecting diverse IoT-based cyber threats. NF-CSE-CIC-IDS2018 contains 8,392,401 flows, of which 12.14% are attack samples, encompassing various contemporary attack scenarios. NF-UNSW-NB15-v3 includes 2,390,275 flows, with 3.98% representing attack traffic, categorized into nine distinct attack types. These datasets provide a comprehensive foundation for training and evaluating NIDS models across a spectrum of attack vectors and network environments.

## 4.2 Data pre-processing and feature engineering

Data preprocessing is a crucial step in any machine learning pipeline. It ensures that the data is clean, well-structured, and ready for analysis. In our work, the first step involves loading the dataset from a CSV file and checking for unnecessary columns. The IPV4_SRC_ADDR, IPV4_DST_ADDR, and Attack columns are dropped if they exist, as they may not contribute valuable information for the model, particularly we want to focus on raw network flow data. Next, categorical data is encoded. The LabelEncoder transforms the categorical 'Label' column into numerical values, allowing the model to process these labels effectively. This step is important because many machine learning algorithms work with numerical data and cannot directly handle categorical variables. Missing values are then filled with zeros, ensuring that the dataset remains complete and preventing errors during model training.

After studying different datasets from literature, we have observed in our primary dataset these features should be included in order to make data more understandable for agent(proposed model) for accurate learning shown in Table 3. Feature engineering enhances the dataset by creating new features that can provide additional insights and improve model performance. The code introduces several new features that are calculated based on existing data, which can help the model capture underlying patterns more effectively.

- Byte-to-Packet Ratios: The features IN BYTE PER_PKT and OUT_BYTE_PER_PKT are computed to reflect the average bytes per packet for incoming and outgoing traffic. This can help the model understand the efficiency of data transmission.

- Packet Rate per Flow Duration: The packet rate features (IN_PKT_RATE and OUT_PKT_RATE) indicate how many packets are transmitted per unit of flow duration, giving insight into network traffic intensity.

- Total Packets and Bytes: Features like TOTAL_PKTS and TOTAL_BYTES aggregate incoming and outgoing packets and bytes, respectively. These summaries are essential for understanding overall network activity.

- Flow Duration per Packet: FLOW_DUR_PER_PKT provides insights into the time taken per packet, which can be indicative of network congestion or performance issues.

- TCP Flag Features: Extracting flags such as TCP_SYN, TCP_ACK, and TCP_FIN helps the model understand the state of TCP connections, which is crucial for network behavior analysis.

- Binary Features for Common Protocols: Features like IS_DNS, IS_HTTP, and IS_HTTPS identify common Layer 7 protocols, allowing the model to focus on relevant traffic types.

- Interaction Features: The ratios IN_OUT_BYTES_RATIO and IN_OUT_PKTS_RATIO provide insights into the relationship between incoming and outgoing traffic.

The features used for network intrusion detection include: **IN_BYTE_PER_PKT**, which is the average bytes per incoming packet; low values may indicate probing or abnormal traffic. **OUT_BYTE_PER_PKT** is the average bytes per outgoing packet, useful for detecting frequent small packets that may suggest malicious activity. **IN_PKT_RATE** measures the incoming packet rate per flow, where high rates can signal scanning or DoS attacks. **OUT_PKT_RATE** is the outgoing

packet rate per flow, with rapid outflow possibly indicating data exfiltration. **TOTAL_PKTS** counts total packets, and spikes can indicate ongoing attacks. **TOTAL_BYTES** counts total bytes, helping establish a baseline for anomaly detection. **FLOW_DUR_PER_PKT** is the average flow duration per packet; low values may indicate rapid-fire attacks.

TCP flags are also considered: **TCP_SYN** indicates SYN flags, where high counts without ACKs may suggest SYN flood attacks; **TCP_ACK** indicates ACK flags, which help detect abnormal connection behavior; **TCP_FIN** indicates FIN flags, useful for detecting irregular connection terminations.

Protocol indicators include **IS_DNS** for DNS traffic (port 53), critical for spotting DNS attacks; **IS_HTTP** for HTTP traffic (port 80), which identifies potential HTTP-based attacks; and **IS_HTTPS** for HTTPS traffic (port 443), detecting attacks even in encrypted traffic.

Finally, traffic ratios are included: **IN_OUT_BYTES_RATIO**, the ratio of incoming to outgoing bytes, where imbalances may indicate data exfiltration, and **IN_OUT_PKTS_RATIO**, the ratio of incoming to outgoing packets, which helps spot unusual traffic patterns.

These features added in this work not only enrich the dataset but also provide essential context for understanding network behavior. In the domain of network intrusion detection, being able to quantify and analyze these dimensions of network traffic is critical for:

**Enhancing Detection Accuracy:** More informative features lead to better model performance by improving the classifier's ability to differentiate between benign and malicious traffic.

**Reducing False Positives:** With a more nuanced understanding of typical network behavior, the model is less likely to misclassify benign activities as threats, which is vital for operational efficiency.

**Improving Response Time:** By identifying specific patterns associated with attacks, security teams can respond more swiftly and effectively to potential threats.

Each added feature serves to enhance the model's understanding of normal and abnormal network behavior, thus facilitating the accurate detection of intrusions and ensuring the security of networked systems.

To address the class imbalance problem commonly encountered in intrusion detection datasets, we employed four advanced synthetic data generation techniques: ADASYN, Borderline-SMOTE, SMOTE-ENN, and K-means SMOTE. ADASYN adaptively generates synthetic samples for minority classes based on the density of difficult-to-learn instances. Borderline-SMOTE focuses on samples near the decision boundary, improving classifier sensitivity to ambiguous cases. SMOTE-ENN combines over-sampling with cleaning via Edited Nearest Neighbors to remove noisy or misclassified samples. K-means SMOTE first clusters data using K-means and then applies SMOTE within each cluster, ensuring more structured and representative sample generation. After that, the dataset is split into training and testing sets using train_test_split. This ensures that the model is evaluated on a separate set of data, which is crucial for assessing its performance reliably. The new class distribution (shown in Table 3) is printed to confirm that the imbalance has been effectively mitigated.

### 4.3 Proposed deep reinforcement learning framework

In this work, a DRL-based NIDS is developed using a Echo State Network architecture that combines dense and ESN layers. The ESN architecture is used as the Agent in the DRL framework, where the agent is trained to detect intrusions by interacting with network data in a sequential decision-making process.

**4.3.1 Agent design and development.** The agent design involves developing a architecture that integrates Dense and ESN layers shown in Algorithm 2. Below is a detailed explanation of each component of the architecture along with the relevant formulas.

**4.3.2 Echo state network as agent.** The Echo State Network (ESN) layer developed for a Deep Reinforcement Learning (DRL) based Network Intrusion Detection System. The ESN layer is implemented as a custom TensorFlow layer that processes sequential input data to capture temporal dependencies, which are crucial for identifying anomalous

patterns in network traffic. The ESN layer is initialized by defining three key parameters: the number of reservoir units (`units`), the spectral radius (`spectral_radius`), and the leaky integration rate (`leaky`). The spectral radius controls the dynamical behavior of the reservoir by adjusting the magnitude of the internal connections, ensuring the echo state property, which is essential for stability. The leaky rate governs the speed at which the reservoir state updates, allowing control over memory depth. During the `build` phase, the input-to-reservoir weight matrix $\mathbf{W}_{in}$ is initialized using the Glorot uniform distribution and is made trainable to allow adaptation during learning. The reservoir recurrent weight matrix $\mathbf{W}$ is initialized randomly and then scaled such that its spectral radius matches the desired value. This reservoir matrix is non-trainable, following the classical ESN principle where only the output layer is typically trained. An initial reservoir state vector $\mathbf{x}$ is also created, initialized to zeros. In the `call` method, the ESN processes input sequences over time. For each sequence in the batch, at each timestep, the reservoir state $\mathbf{x}$ is updated based on the current input and the previous state. The update rule combines the previous state and a non-linear transformation of the input and recurrent contributions, regulated by the leaky parameter. The final output of the layer is the last reservoir state after processing the entire sequence, which can then be used by the DRL agent for decision-making. During initialization of the reservoir weight matrix $\mathbf{W}$, the matrix is normalized to ensure the desired spectral radius:

$$\mathbf{W} \leftarrow \frac{\text{spectral\_radius}}{\rho(\mathbf{W})} \times \mathbf{W} \tag{1}$$

where $\rho(\mathbf{W})$ denotes the spectral radius, defined as the maximum absolute eigenvalue of $\mathbf{W}$. At each timestep $t$, the reservoir state is updated using the following leaky integration formula:

$$\mathbf{x}(t) = (1 - \alpha)\mathbf{x}(t-1) + \alpha \tanh\left(\mathbf{W}_{in}^{\top}\mathbf{u}(t) + \mathbf{W}^{\top}\mathbf{x}(t-1)\right) \tag{2}$$

where:

- $\alpha$ is the leaky integration rate,

- $\mathbf{u}(t)$ is the input vector at time $t$,

- $\mathbf{W}_{in}$ is the input-to-reservoir weight matrix,

- $\mathbf{W}$ is the internal reservoir weight matrix,

- $\mathbf{x}(t-1)$ is the previous reservoir state.

**Algorithm 1 Echo State Network (ESN) Layer Forward Pass**

```
 1:  Class ESNLayer(units, input_dim, α)
 2:  Initialize W_in, W, zero state
 3:  function FORWARD(U ∈ ℝ^{B×T×D})
 4:      X_final ← []
 5:      for all batch item u_b ∈ U do
 6:          x ← 0 ∈ ℝ^{units}
 7:          for all timestep u_b^t ∈ u_b do
 8:              a ← W_in^⊤ u_b^t + W^⊤ x
 9:              x ← (1 − α) · x + α · tanh(a)
10:          end for
11:          Append x to X_final
12:      end for
13:      return X_final
14: end function
```

After processing all time steps, the output of the ESN layer is the final state of the reservoir:

$$\mathbf{y} = \mathbf{x}(T) \tag{3}$$

where $T$ is the last timestep of the sequence. The ESN layer is an efficient temporal feature extractor that provides dynamic memory to the DRL agent operating in a NIDS framework. By utilizing a fixed reservoir with a controlled spectral radius and a trainable input mapping, the ESN enables robust sequence modeling while maintaining computational efficiency, which is crucial for real-time network anomaly detection systems.

**Agent Architecture:** The agent begins with a fully connected Dense layer comprising 128 neurons. This layer performs a linear transformation followed by a non-linear activation. Mathematically, the output $y$ of the Dense layer is given by:

$$y = \text{ReLU}(Wx + b)$$

where $W \in \mathbb{R}^{128 \times \text{state\_size}}$ is the weight matrix, $b$ is the bias vector, and $x$ is the input feature vector. The ReLU activation function is defined as:

$$\text{ReLU}(z) = \max(0, z)$$

which ensures non-linearity and sparsity in the activations, helping the model to learn complex feature representations.

Following the Dense layer, a Dropout layer is applied with a dropout rate of 0.1. Dropout is a regularization technique where, during training, each neuron is independently set to zero with probability $p = 0.1$. This prevents overfitting by introducing noise during learning. If $h$ denotes the activations before dropout, the activations after applying dropout, $\tilde{h}$, are given by:

$$\tilde{h}_i = \begin{cases} 0 & \text{with probability } 0.1 \\ \frac{h_i}{0.9} & \text{with probability } 0.9 \end{cases}$$

where scaling by $\frac{1}{1-p}$ ensures that the expected value remains unchanged at test time. After dropout, the output tensor is reshaped from a flat shape (128,) to (1, 128). This operation prepares the data for processing by sequential layers such as the ESN layers, which expect time series data. The reshaping operation does not change the values, only their arrangement in memory.

An ESN layer is then applied, as previously described. After the first ESN layer, another reshaping step is performed to maintain the temporal structure for further processing by the next ESN layer. This second ESN layer continues to extract temporal features from the processed sequence. Following the ESN layers, the agent uses another Dense layer with 128 neurons and ReLU activation. This layer further transforms the high-level temporal features extracted by the ESNs into a richer latent space, using the same mathematical form as earlier:

$$y = \text{ReLU}(Wx + b)$$

to enhance the model's ability to classify complex patterns. Next, a Dropout layer with a rate of 0.1 is again applied to reduce the risk of overfitting. Afterward, the agent incorporates a Dense layer with 100 neurons, ReLU activation, and L2 regularization. L2 regularization adds a penalty proportional to the square of the magnitude of the weights. The loss function $L$ is augmented as:

$$L_{\text{new}} = L_{\text{original}} + \lambda \|W\|_2^2$$

where $\lambda = 0.01$ is the regularization coefficient, and $\|W\|_2^2$ denotes the sum of squares of all the weight parameters. This discourages the model from learning excessively large weights. A Dropout layer with rate 0.1 follows again, maintaining consistency in regularization after dense transformations.

Subsequently, another Dense layer with 100 neurons, ReLU activation, and L2 regularization (with the same penalty term as above) is included. This layer further refines the features before moving to a compressed lower-dimensional space. The next Dense layer has 32 neurons and uses ReLU activation. This layer acts as a bottleneck, reducing the dimension of the feature representation and encouraging the agent to learn compact and efficient encodings of the input data.

Finally, the agent ends with a Dense output layer consisting of 2 neurons, corresponding to the number of classes (for instance, attack and benign traffic). The activation function used here is the softmax, which produces a probability distribution over the classes. The softmax function for class $i$ given logits $z_i$ is defined as:

$$\text{softmax}(z_i) = \frac{e^{z_i}}{\sum_j e^{z_j}}$$

ensuring that the output values are positive and sum to 1.

The agent is compiled using categorical crossentropy as the loss function, which is appropriate for multi-class classification tasks. The categorical crossentropy loss for a single sample is given by:

$$\mathcal{L} = -\sum_i y_i \log(\hat{y}_i)$$

where $y_i$ is the true label (one-hot encoded) and $\hat{y}_i$ is the predicted probability for class $i$. The optimizer selected is Adam, a stochastic gradient-based optimization method that adapts learning rates individually for each parameter. The model's performance is evaluated using accuracy as the primary and weighted metric. This architectural design efficiently combines spatial feature extraction (Dense layers), regularization (Dropout and L2 penalties), temporal feature learning (ESN layers), and probabilistic classification (Softmax output) to form a powerful and robust agent for DRL-based network intrusion detection. The model is trained with a state size equal to the input feature dimension, spectral radius $\rho = 0.9$, leaky rate $\alpha = 1.0$, reservoir sizes of 128 and 256 units, batch size 32 and 31 training episodes.

**Algorithm 2 Agent Construction**

```
 1:  Class Agent(state_size)
 2:  function BUILDMODEL
 3:      Initialize Sequential model
 4:      Add Dense layer (128 units, ReLU, input shape = state_size)
 5:      Add Dropout (rate = 0.1)
 6:      Reshape to (1,128)
 7:      Add ESN layer (128 units, ρ = 0.9, α = 0.1)
 8:      Reshape to (1,128)
 9:      Add ESN layer (128 units, ρ = 0.9, α = 0.1)
10:      Add Dense layer (128 units, ReLU)
11:      Add Dropout (rate = 0.1)
12:      Add Dense layer (100 units, ReLU, L2=0.01)
13:      Add Dropout (rate = 0.1)
14:      Add Dense layer (100 units, ReLU, L2=0.01)
15:      Add Dense layer (32 units, ReLU)
16:      Add Dense layer (2 units, Softmax)
17:      Compile model (loss = categorical crossentropy, optimizer = Adam, metrics = accuracy)
18:      return model
19:  end function
```

**4.3.3 Theoratical justification of using ESN over LSTM/GRU.** The choice of Echo State Networks (ESNs) over fully trainable recurrent architectures such as LSTM and GRU is theoretically motivated by stability, efficiency, and suitability for online decision-making in DRL-based NIDS. ESNs rely on a fixed, randomly connected reservoir with a controlled spectral radius, which guarantees the echo state property and ensures stable, contractive dynamics without the risk of exploding or vanishing gradients inherent in backpropagation through time used by LSTM/GRU. The leaky integration mechanism provides an explicit and interpretable control over memory depth, allowing the reservoir to capture multi-scale temporal dependencies critical for network traffic analysis. Since only the input mapping and downstream layers are trained, ESNs significantly reduce the number of trainable parameters and computational overhead, enabling faster convergence and improved robustness in non-stationary, real-time environments. In contrast, LSTM/GRU models require extensive gradient-based optimization over recurrent weights, making them more prone to overfitting and less stable under rapidly changing traffic patterns. Therefore, ESNs offer a theoretically grounded trade-off between expressive temporal modeling and computational efficiency, making them particularly well-suited for DRL agents operating in high-throughput network intrusion detection systems.

**4.3.4 DRL training.** The DRL agent used here is based on Deep Q-Network (DQN) principles, but with critical enhancements. The agent employs Echo State Network layers instead of recurrent layers like LSTM or GRU to better capture temporal dependencies.

The agent interacts with the environment to learn an optimal policy that maximizes cumulative rewards over time.

The key mathematical components are as follows:

- **Q-learning Update Rule**:

$$Q(s_t, a_t) \leftarrow Q(s_t, a_t) + \alpha \left( r_t + \gamma \max_{a'} Q(s_{t+1}, a') - Q(s_t, a_t) \right) \tag{4}$$

where:

- $s_t$: Current state
- $a_t$: Action taken
- $r_t$: Immediate reward
- $\gamma$: Discount factor
- $\alpha$: Learning rate (implicitly handled by optimizer)

- **Action Selection via $\epsilon$-greedy Policy**:

$$a_t = \begin{cases} \text{random action} & \text{with probability } \epsilon \\ \arg\max_a Q(s_t, a) & \text{with probability } 1 - \epsilon \end{cases} \tag{5}$$

- **Epsilon Decay for Exploration-Exploitation Tradeoff**:

$$\epsilon = \max(\epsilon_{\min}, \epsilon \times \epsilon_{\text{decay}}) \tag{6}$$

- **Discounted Reward Calculation**:

$$G_t = \sum_{k=0}^{T-t} \gamma^k r_{t+k} \tag{7}$$

The agent is compiled with the categorical cross-entropy loss function and the Adam optimizer with a learning rate of $10^{-4}$. The training procedure consists of two main parts: **Interaction and Memory Storage** and **Experience Replay and Model Update**. Below are the pseudocodes that describe the full training setup.

### Algorithm 3 Interaction and Memory Storage

```
1:   Initialize agent with state size and action size
2:   for each episode e ∈ {1, 2, ..., EPISODES} do
3:      Reset environment, get initial state s_0
4:      Initialize cumulative reward R ← 0
5:      Initialize discounted reward G ← 0
6:      for each time step t do
7:         Select action a_t using ε-greedy policy
8:         Execute a_t, observe reward r_t and next state s_{t+1}
9:         Store (s_t, a_t, r_t, s_{t+1}, done) in memory
10:        R ← R + r_t
11:        G ← G + r_t · γ^t
12:        Transition to s_{t+1}
13:        if done then
14:           break
15:        end if
16:     end for
17:     Record performance metrics
18: end for
```

### Algorithm 4 Experience Replay and Model Update

```
1:   function ExperienceReplayUpdate(memory, batch_size, Q, γ, ε, ε_decay)
2:      if size(memory) > batch_size then
3:         Sample random mini-batch of experiences from memory
4:         for all (s, a, r, s', done) in mini-batch do
5:            if done then
6:               y ← r
7:            else
8:               y ← r + γ max_{a'} Q(s', a')
9:            end if
10:           Predict current Q-value: Q(s, a)
11:           Update Q(s, a) towards target y
12:           Train model on updated targets (with class weights if applicable)
13:        end for
14:        Update exploration parameter: ε ← ε × ε_decay
15:     end if
16: end function
```

The agent periodically saves its model weights every 10 episodes and computes classification metrics including the confusion matrix and classification report.

Performance tracking includes:

- Total rewards accumulated

- Average rewards per episode

- Discounted rewards

- Time complexity per episode

- Convergence rate (difference in average rewards between episodes)

  

At every 10 episodes, the model's classification ability is evaluated on the test set by predicting labels and calculating performance metrics.

Key Parameters and Hyperparameters:

- Discount factor $\gamma$ = 0.96

- Initial exploration rate $\epsilon$ = 1.0

- Minimum exploration rate $\epsilon_{min}$ = 0.07

- Exploration decay factor $\epsilon_{decay}$ = 0.999

- Learning rate $\alpha$ = 0.0001

The overall training continues until convergence is observed, evaluated by convergence rate and stabilization of performance metrics.

**4.3.5 Reward mechanism.** The reward mechanism in the agent is based on the reinforcement learning, where the agent interacts with the environment, takes actions, and receives rewards based on its actions. The mechanism is explained in the following steps.

1. **Reward Assignment (during the act method)** The agent selects an action at each timestep in the environment using an $\epsilon$-greedy policy. After taking an action, the agent receives a **reward** based on the correctness of the action. The reward mechanism is defined as follows:

$$r_t = \begin{cases} 1 & \text{if the action } a_t \text{ is correct} \\ -1 & \text{if the action } a_t \text{ is incorrect} \end{cases}$$

This reward mechanism means the agent is rewarded with a positive value (+1) for taking a correct action and penalized with a negative value (−1) for taking an incorrect action.

2. **Cumulative Reward Calculation** At each timestep, the agent accumulates the rewards into a **total reward**. The cumulative reward over the entire episode is computed as:

$$\text{total\_reward} = \sum_{t=0}^{T} r_t$$

where $T$ is the total number of timesteps in the episode, and $r_t$ is the reward obtained at each timestep $t$.

The agent also keeps track of **discounted rewards**:

$$G_t = \sum_{k=0}^{T-t} \gamma^k r_{t+k}$$

Here, $G_t$ represents the discounted reward starting from timestep $t$, $r_{t+k}$ is the reward received at timestep $t+k$, and $\gamma$ is the **discount factor** (set to 0.96 in this case), which controls how much importance is given to future rewards. The higher the value of $\gamma$, the more the agent considers future rewards as part of the current reward.

3. **Reward Update in Memory (during the remember method)** The agent stores its experience (state, action, reward, next state, and whether the episode is done) in its **experience replay memory**. This is important for learning from past experiences, leading to better sample efficiency. The memory stores experiences in the form of:

$$\text{Memory} = \{(s_t, a_t, r_t, s_{t+1}, \text{done})\}$$

where: $-s_t$ is the current state at timestep $t$, $-a_t$ is the action taken by *t*he agent at timestep $t$, $-r_t$ is the immediate reward received by the agent after taking action $a_t$, $-s_{t+1}$ is *t*he next state the agent transitions to after taking the action $a_t$, $-\text{done}$ is a boolean flag indicating whether the episode has ended.

4. **Reward Update in the replay Method** The agent uses **experience replay** to learn from previously stored experiences. During training, the agent samples a mini-batch of experiences from its memory and updates its Q-values based on the rewards. The Q-learning update rule is applied to compute the **target value** $y$ for each experience:

$$y = \begin{cases} r_t & \text{if episode ends} \\ r_t + \gamma \max_{a'} Q(s_{t+1}, a') & \text{if episode continues} \end{cases}$$

Here:

- When the episode ends ('done = True'), the agent only uses the immediate reward $r_t$ as the target for the Q-value update.

- When the episode continues ('done = False'), the agent calculates the future reward using the **Bellman equation**, where it adds the discounted maximum future reward $\gamma \max_{a'} Q(s_{t+1}, a')$ from the next state to the current reward $r_t$.

**4.3.6 Convergence criterion.** The DRL agent was trained for a fixed number of episodes (EPISODES = 31), determined through empirical analysis of training stability. Convergence was evaluated by monitoring the moving average of rewards per episode and classification accuracy on a held-out validation set. Training dynamics consistently showed that both metrics plateaued after approximately 15–20 episodes, with variations of less than 1% in the final 10 episodes. The chosen episode count provides a conservative buffer beyond this observed stabilization point, ensuring robust policy convergence without overfitting.

**4.3.7 Hardware specification.** To ensure the effective implementation of the proposed DRL-based network intrusion detection system, specific hardware requirements are essential shown in Table 2. Below is a detailed breakdown of the hardware used for experimentation:

Processor: The system was tested using an Intel® Core™ i5-8265U CPU with a base clock speed of 1.60 GHz and 8 logical cores. This processor offers enough computational power to handle the training and inference of the deep neural networks (DNN) and Long Short-Term Memory (LSTM) models, though more powerful CPUs would reduce the overall training time and allow for better real-time performance in larger network environments.

RAM: A total of 12 GB of RAM was utilized, which proved sufficient for managing network traffic data and deep learning model training. However, increasing the RAM would be beneficial for handling larger datasets or for parallel processing during model training and validation.

Graphics: The system was equipped with Intel® UHD Graphics 620, which includes 6197 MB of total graphics memory. Out of this, 128 MB is dedicated display memory, while 6069 MB is shared memory. Although integrated graphics were sufficient for the experiments conducted in this study, the use of dedicated GPUs (e.g., NVIDIA) with CUDA support is highly recommended for faster model training and handling of large-scale network intrusion datasets.

**Table 2. Hardware requirements for DRL-based intrusion detection.**

| Component | Specification | Remarks |
|---|---|---|
| Processor | Intel® Core™ i5-8265U CPU @ 1.60GHz (8 logical cores) | Sufficient for model training and inference, though more powerful CPUs can reduce training time. |
| RAM | 12 GB | Adequate for network traffic data and model training. More RAM is recommended for larger datasets. |
| Graphics | Intel® UHD Graphics 620 | Integrated graphics were sufficient, but a dedicated GPU is recommended for better performance. |
| Total Graphics Memory | 6197 MB | Total memory available for graphics processing, shared between system and display. |
| Display Memory | 128 MB | Dedicated display memory for handling visual output. |
| Shared Memory | 6069 MB | Shared between system and graphics tasks, reducing GPU burden. |

While the above hardware was adequate for experimental purposes, scaling the system for real-time deployment in larger network environments might require more advanced hardware, especially dedicated GPUs for efficient deep learning processing and larger memory for handling massive traffic data streams.

## 5 Results analysis

After successful training process, author has conducted multi-dataset validation, Comparative Analysis on Synthetic Data balancing methods, used LIME for understanding the feature contribution and tested the Adaptive modeling functionality on our proposed model (agent). Author has used performance matrix such as accuracy, f1-score, recall, precision, AUC-ROC, total reward, average reward, discounted reward and confusion matrix for evaluating the proposed solution shown in Table 6. **All the datasets were trained like provided Table 3 training-testing strategy format.**

### 5.1 Performance analysis

In Tables 4, 5, we can see the classification report of both Training and Testing of our network intrusion detector. Tables 4–6 present a comprehensive evaluation of the proposed DRL-based network intrusion detection system using the NF-BoT-IoT dataset. Table 4 shows the classification report on training data at episode 30, where both classes (benign = 0 and attack = 1) achieved high precision, recall, and F1-scores (all approximately 0.99), indicating excellent model learning during training.

| Metric | Score |
|---|---|
| Convergence Rate | 0.2000 |
| Total Reward | 34 |
| Average Reward | 0.68 |
| Discounted Reward | 15.33 |

Similarly, Table 5 illustrates that the model maintained this high performance on unseen testing data, reflecting strong generalization. Table 6 further confirms these findings with nearly identical train and test scores across AUC-ROC, accuracy, precision, recall, and F1-score, all around 0.987. Lastly, the reward metrics table reveals a total reward of 34, average reward of 0.68, a convergence rate of 0.2000, and a discounted reward of 15.33, demonstrating stable and steadily improving reinforcement learning behavior throughout training.

**5.1.1 Comparison of synthetic methods.** While dealing with class imbalance issue we have used several methods like SMOTE-ENN, ADYSN, Kmeans SMOTE and Borderline SMOTE. The impact of choosing right method for accurate intrusion detection is important. A set of experiment conducted with following synthetic data generation methods

**Table 3. Training-testing data split strategy.**

| Dataset | Data Present in Trainset | Not Present in Trainset |
|---|---|---|
| NF-BoT-IoT | Benign, Reconnaissance and DDoS | DoS |
| NF-ToN-IoT | Benign, DDoS, Injection, MITM, Password and XSS | Scanning, DoS, Backdoor and Ransomware |
| NF-ToN-IoT-v2 | Benign, DDoS, Injection, MITM, Password and XSS | Scanning, DoS, Backdoor and Ransomware |
| NF-UNSW-NB15 | Benign, Fuzzers, Analysis, Exploits, Generic, Reconnaissance, Shellcode and Worms | DoS and Backdoor. |
| NF-CSE-CIC-IDS 2018 | Benign, Infilteration, Bot, Brute Force -Web, Brute Force -XSS, DDOS attack-HOIC, SQL Injection, DDoS attacks-LOIC-HTTP, DDOS attack-LOIC-UDP,DoS attacks-Slowloris, FTP-BruteForce | DoS attacks-GoldenEye, DoS attacks-Hulk, DoS attacks-SlowHTTPTest and SSH-Bruteforce |
| NF-UNSW-NB15-v3 | Benign, Fuzzers, Exploits, Reconnaissance, Generic, Shellcode, Analysis, Worms | DoS, and Backdoor |

**Table 4. Classification report on TRAINING data at episode 30, dataset:NF-BoT-IoT (Weighted).**

| Class | Precision | Recall | F1 Score | Support |
|---|---|---|---|---|
| 0 | 0.99 | 0.98 | 0.99 | 468736 |
| 1 | 0.98 | 0.99 | 0.99 | 469250 |
| **ACCURACY** | | | **0.99** | **937986** |
| **MACRO AVG** | **0.99** | **0.99** | **0.99** | **937986** |
| **WEIGHTED AVG** | **0.99** | **0.99** | **0.99** | **937986** |

**Table 5. Classification report on TESTING data, dataset:NF-BoT-IoT (Weighted).**

| Class | Precision | Recall | F1 Score | Support |
|---|---|---|---|---|
| 0 | 0.99 | 0.98 | 0.99 | 117506 |
| 1 | 0.98 | 0.99 | 0.99 | 116991 |
| **ACCURACY** | | | **0.99** | **234497** |
| **MACRO AVG** | **0.99** | **0.99** | **0.99** | **234497** |
| **WEIGHTED AVG** | **0.99** | **0.99** | **0.99** | **234497** |

shown in Table 7. Kmeans SMOTE consistently outperformed the other methods across all metrics, making it the most effective approach in handling class imbalance for network intrusion detection. Meanwhile, SMOTE-ENN had the lowest performance, suggesting it might not be the ideal choice in this context.

**5.1.2 Multi-data validation.** In order to prove robustness of proposed approach in various type of dataset, we have conducted multi-dataset validation strategy shown in Table 8. Table 8 presents the performance of the proposed method validated on multiple network flow datasets, namely NF-BoT-IoT, NF-UNSW-NB15, NF-ToN-IoT, NF-ToN-IoT-v2, NF-CSE-CIC-IDS2018, and NF-UNSW-NB15-v3. The evaluation metrics include Accuracy, Precision, Recall, F1 Score, and AUC-ROC. Among these datasets, NF-UNSW-NB15-v3 achieved the highest overall performance with an accuracy, recall, F1 score, and AUC-ROC of 0.9885, indicating balanced and strong classification capability. The NF-BoT-IoT and NF-UNSW-NB15 datasets also performed well, with high scores across all metrics, demonstrating the model's generalizability across attack scenarios. In contrast, NF-ToN-IoT-v2 showed the lowest performance, especially in terms of accuracy (0.9266)

**Table 6. Performance metrics of DRL-based network intrusion detector, dataset:NF-BoT-IoT.**

| Metric | Train Data | Test Data |
|---|---|---|
| AUC-ROC | 0.9872 | 0.9874 |
| Accuracy | 0.9872 | 0.9874 |
| Precision | 0.9842 | 0.9839 |
| Recall | 0.9903 | 0.9908 |
| F1-Score | 0.9872 | 0.9874 |

**Table 7. Performance metrics of different synthetic data methods.**

| Method | Accuracy | Precision | Recall | F1 Score | AUC-ROC |
|---|---|---|---|---|---|
| ADASYN | 0.9452 | 0.9643 | 0.9246 | 0.9440 | 0.9452 |
| B-SMOTE | 0.9458 | 0.9682 | 0.9216 | 0.9443 | 0.9457 |
| SMOTE-ENN | 0.8817 | 0.9116 | 0.8476 | 0.8784 | 0.8820 |
| Kmeans SMOTE | 0.9874 | 0.9839 | 0.9908 | 0.9874 | 0.9874 |

**Table 8. Multi data validation.**

| Method | Accuracy | Precision | Recall | F1 Score | AUC-ROC |
|---|---|---|---|---|---|
| NF-BoT-IoT | 0.9874 | 0.9839 | 0.9908 | 0.9874 | 0.9874 |
| NF-UNSW-NB15 | 0.9778 | 0.9994 | 0.9563 | 0.9773 | 0.9778 |
| NF-ToN-IoT | 0.9529 | 0.9138 | 0.9999 | 0.9549 | 0.9530 |
| NF-ToN-IoT-v2 | 0.9266 | 0.9040 | 0.9546 | 0.9286 | 0.9266 |
| NF-CSE-CIC-IDS2018 | 0.9731 | 0.9986 | 0.9476 | 0.9724 | 0.9731 |
| NF-UNSW-NB15-v3 | 0.9885 | 0.9852 | 0.9918 | 0.9885 | 0.9885 |

and precision (0.9040), suggesting slightly less effectiveness on this dataset. Overall, the results highlight the robustness of the model across diverse network intrusion detection environments.

**5.1.3 Prior study comparison.** Table 9 presents a performance comparison between the proposed ESN model and various existing deep learning architectures on the NF-BoT-IoT dataset. Although the LSTM+DNN model slightly outperforms in terms of accuracy and recall, the ESN model achieves nearly equivalent performance with an accuracy of 0.9874, precision of 0.9839, recall of 0.9908, F1-score of 0.9874, and AUC-ROC of 0.9874. Compared to more complex models like GRU+LSTM and CNN+LSTM, the ESN not only provides better or comparable results but does so with significantly lower computational complexity due to its lightweight design. This highlights ESN's effectiveness in achieving high accuracy while being computationally efficient, making it a suitable choice for real-time intrusion detection systems with constrained resources.

The comparative results presented in Table 9 demonstrate that the proposed ESN achieves a highly competitive detection performance while maintaining significantly lower computational complexity than gated recurrent architectures. Although the LSTM+DNN model attains the highest accuracy of 0.9895, it does so at the expense of a substantially higher computational cost, requiring approximately $4.34 \times 10^5$ floating-point operations (FLOPs) per inference shown in Table 10. In contrast, the proposed ESN achieves an accuracy and F1-score of 0.9874 with only $1.63 \times 10^5$ FLOPs, corresponding to a reduction of more than 2.6× in computational cost compared to the LSTM+DNN baseline. This efficiency arises from the fixed and sparse reservoir dynamics of the ESN, which eliminate backpropagation through time and reduce the number of floating-point operations required for temporal modeling. Moreover, when compared with GRU+LSTM and CNN+LSTM models, the ESN attains comparable or superior F1-score and AUC-ROC values while requiring fewer computational

**Table 9. Prior study comparison.**

| Model | Accuracy | Precision | Recall | F1 Score | AUC-ROC |
|---|---|---|---|---|---|
| GRU+LSTM [51] | 0.9678 | 0.9940 | 0.9411 | 0.9668 | 0.9677 |
| CNN+LSTM [52,53] | 0.9556 | 0.9523 | 0.9346 | 0.9534 | 0.9556 |
| DNN [54] | 0.8845 | 0.9961 | 0.7715 | 0.8696 | 0.8843 |
| LSTM+DNN [55] | 0.9895 | 0.9794 | 1.0000 | 0.9896 | 0.9895 |
| **ESN (Proposed)** | 0.9874 | 0.9839 | 0.9908 | 0.9874 | 0.9874 |

**Table 10. Computational cost comparison in terms of FLOPs per inference; Dataset: NF-BoT-IoT.**

| Model | FLOPs (per sample) |
|---|---|
| CNN+LSTM | $1.06 \times 10^5$ |
| DNN | $1.39 \times 10^5$ |
| GRU+LSTM | $1.79 \times 10^5$ |
| **ESN (Proposed)** | $1.63 \times 10^5$ |
| LSTM+DNN | $4.34 \times 10^5$ |

resources, highlighting its favorable trade-off between detection performance and computational efficiency. These characteristics make the proposed ESN particularly suitable for real-time and resource-constrained intrusion detection systems. All FLOPs are computed for a single forward pass with an input state dimension of 27 features.

## 5.2 Interpreting feature contributions with LIME

LIME (Local Interpretable Model-agnostic Explanations) is a technique used to interpret the predictions of machine learning models, especially complex and black-box models like deep learning networks or ensemble models. It helps explain how individual features contribute to the prediction of a specific instance, making the model's decision-making process more transparent.

The LIME output visualization shown in Fig 1 provides a detailed breakdown of how the network intrusion detection model arrived at its prediction for a specific instance. This output provides an explanation of a model's prediction for test instance 136240, where the model predicts the instance as "Benign" with a probability of 1.00 and "Attack" with a probability of 0.00. The explanation identifies the most influential features contributing to the prediction, with the most important factor being the number of incoming packets (`IN_PKTS`) greater than 38.79, followed by other features such as incoming bytes (`IN_BYTES`) greater than 18.67 and the presence of HTTP traffic (`IS_HTTP ≤ −0.03`). Additionally, several other features like outgoing packets, packet rate, TCP flags, and protocol types also influence the prediction, with varying degrees of importance. The model primarily relies on packet-related features and protocol characteristics to classify the instance as benign. Features like Label, `TCP_SYN`, and `L4_SRC_PORT` also play a smaller role in the decision-making process.

## 5.3 Scenario: Adaptive threat detection in a corporate network

Let's consider a scenario in a large corporate environment, where a multinational company manages thousands of interconnected devices, including employee workstations, servers, IoT devices, and mobile endpoints. The company has a high volume of network traffic due to remote work, cloud services, and partner integrations, making it a prime target for cyber threats such as DDoS attacks, malware infiltration, or data exfiltration attempts.

In this scenario, the company's security team uses traditional signature-based IDS. These systems effectively detect known attacks but struggle with identifying novel or evolving threats. For instance, if an advanced persistent threat (APT)

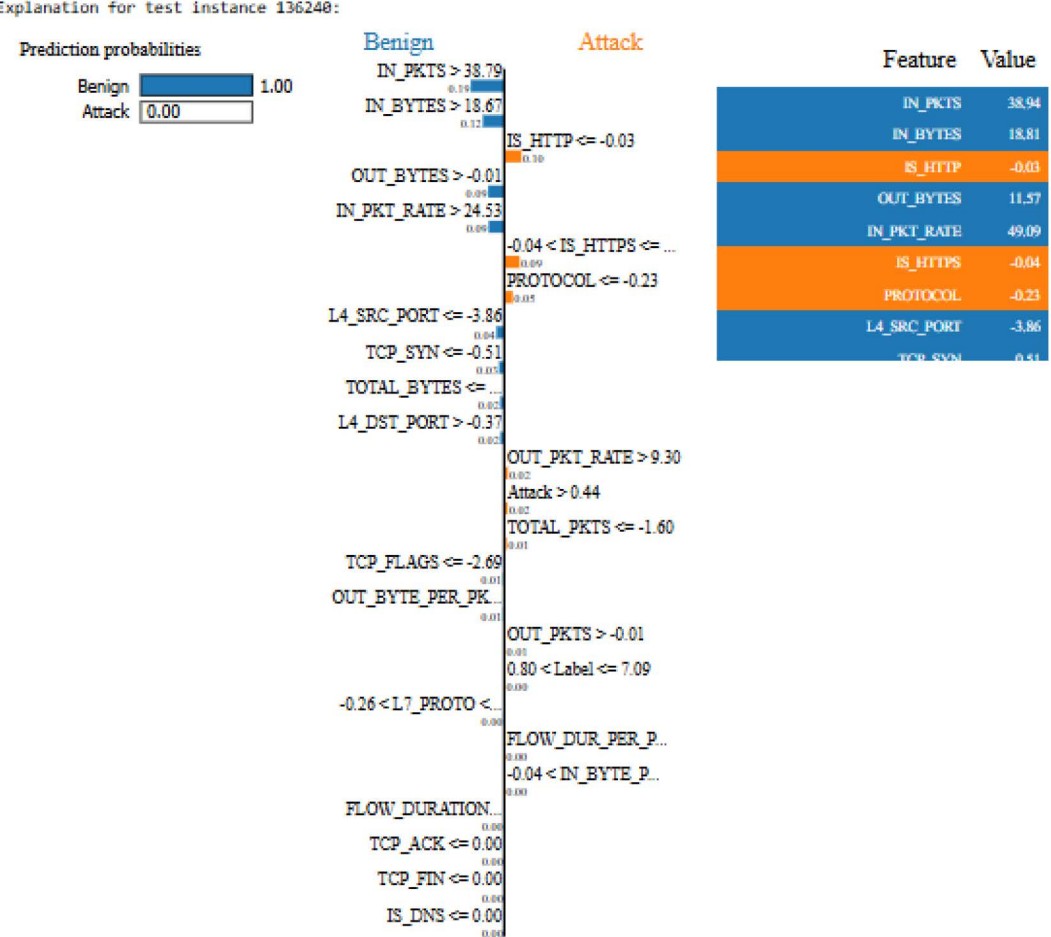

**Fig 1. Feature contribution visualization using LIME.**

actor deploys a new variant of malware or initiates a low-frequency DDoS attack (slow-loris), the static IDS might miss the attack until significant damage is done.

Key Challenges:

- Dynamic Threats: Attackers continuously change tactics, techniques, and procedures (TTPs), making it difficult for static models or signature-based systems to keep up.

- High Volume of Traffic: With the vast amount of network traffic flowing through the corporate network, detecting low-frequency or low-signature attacks (such as slow-loris DDoS or stealthy data exfiltration) becomes challenging.

- Class Imbalance: The network traffic data is heavily skewed towards normal behavior, making it harder for the system to detect rare but critical attack instances.

**5.3.1 Advantages of proposed DRL-based approach.** In this setting, the proposed DRL approach combined with a ESN model offers several advantages over traditional systems:

- Real-Time Detection of Unseen Attacks: DRL continuously learns and adapts to changes in the network environment. For example, if an attacker launches a previously unknown form of data exfiltration or a new variant of ransomware, the

DRL model can adapt by learning from interactions and feedback, adjusting its detection strategies to flag suspicious behavior even when the threat is not part of the pre-trained data.

- Adaptation to Evolving Attack Strategies: By excluding certain attack types (like DoS) from the training phase but including them in the testing phase, the model demonstrates an ability to generalize and adapt. If the corporate network starts facing new attack vectors, our DRL-based system would be able to detect the evolving nature of the attack and prevent the disruption before it becomes catastrophic.

- Handling Imbalanced Data: Network traffic in such environments is overwhelmingly normal with rare but critical intrusion attempts. our approach addresses this by using advanced data balancing techniques like K-means SMOTE. In this scenario, the imbalanced network data (i.e., large volumes of legitimate traffic with few intrusion samples) is handled more effectively, ensuring the model doesn't miss subtle yet critical threats.

- Robust Multi-Dataset Validation: Since the model has been validated across multiple datasets, it can be confidently deployed in environments with different types of network flow data, such as cloud traffic, VPN connections, or IoT communications within the corporate network. This multi-dataset validation ensures that the DRL-based system is resilient to variations in traffic patterns, making it versatile across different sectors (e.g., banking, healthcare, or manufacturing).

**5.3.2 Outcome in the scenario.** The DRL-based model quickly detects an anomalous pattern in an employee's workstation communicating with a previously unused external IP address. Traditional systems, relying on known signatures, would not flag this activity as malicious. The model learns from this event and classifies the behavior as potentially malicious due to patterns of low-volume, periodic data transmissions consistent with data ex filtration attempts. It isolates the device and alerts the security team, who discover an APT actor attempting to extract sensitive corporate data. Additionally, the model notices a slow-buildup of traffic towards a critical server, recognizing a low-frequency DDoS attack in progress. It mitigates the attack by throttling suspicious traffic and preventing server downtime. In this scenario, our DRL-based network intrusion detection system, with its adaptive learning capabilities and robust handling of imbalanced datasets, provides more comprehensive protection compared to traditional methods. It excels at detecting and responding to novel, evolving threats in real-time, which is critical for securing complex and high-traffic corporate networks.

## 5.4 Need for advanced learning depth

In traditional deep learning approaches for network intrusion detection, models are trained to recognize specific attack patterns based on historical data. While these methods can achieve good detection rates, they are inherently static, meaning they cannot easily adapt to new, unknown attack types that evolve over time. The fundamental limitation of such models is their reliance on patterns seen during training. When they encounter an attack that deviates from the learned patterns, they may struggle to detect it, especially if the computational cost is optimized, potentially sacrificing adaptability for speed.

DRL, as used in our proposed model, addresses this limitation by incorporating an adaptive learning mechanism. The key difference lies in DRL's ability to dynamically interact with the environment, learning optimal detection strategies over time rather than relying solely on pre-defined attack signatures. This continuous learning aspect allows DRL-based models to detect previously unseen or novel attack patterns, often referred to as zero-day attacks, where an attack is detected even though it was not present in the training data. In the experiment, specific attack types, such as DoS, were deliberately excluded from the training set but included in the test set. Traditional deep learning methods, when faced with such a setup, often fail to detect these novel attacks effectively, as their detection capabilities are tied to the patterns they have seen during training.

However, our DRL-based approach demonstrates superior performance under these conditions. By interacting with the evolving attack patterns in the test data and continuously refining its strategies, the DRL model shows a marked improvement in detecting these excluded attack types. This ability to generalize and adapt, even in the face of unseen attacks, provides a significant advantage over traditional static models. It ensures that the model can function in real-time network environments, where attacks are constantly evolving and attackers often introduce new techniques that deviate from historical trends.

Thus, while traditional deep learning methods may come with lower computational costs, they sacrifice the crucial ability to adapt in real-time, limiting their effectiveness against zero-day or novel attacks. DRL's adaptive modeling, as evidenced by our zero-day attack detection experiment, justifies the added complexity, as it enhances security in dynamic and unpredictable network environments, providing long-term robustness that traditional methods lack.

## 6 Discussion

The feature engineering and dataset adjustments explained in section 3-B also contribute to the success of this adaptive modeling approach. In this case, by excluding the samples of the DoS attack type for training and saving only the test portion, the experiment has achieved, in fact, checking the model capability for adaptation against unseen data. The high performance, despite these constraints, gives an excellent reinforcement to its generalization. This flexibility is crucial in the dynamic cybersecurity domain for modeling not only the detection of known threats but also for finding out novel attack patterns, which might be used to exploit vulnerabilities. Besides, the total weighted accuracy of 99% across the whole data space underlines the robustness of the model. The high accuracy of results obtained while testing the model against data with previously unseen types of attacks underlined the efficiency of the adaptive modeling strategy. That will keep the model agile and certain for both known and unknown threat detection in real time, which is one main requirement of intrusion detection systems operating in complex and ever-evolving network environments.

## 7 Limitation

While the proposed DRL-based network intrusion detection system offers several advancements over existing solutions, it also has some limitations that should be acknowledged. The integration of deep reinforcement learning, ESN architectures, and advanced data balancing techniques can lead to a complex implementation process. Organizations may require specialized knowledge and skills to effectively deploy and maintain the system. he training phase of deep reinforcement learning involves running numerous simulations and iterations to optimize the model's parameters. This requires substantial computational resources, including powerful GPUs or TPUs, which can be costly. Training deep learning models can take a significant amount of time, especially as the size and complexity of the dataset increase. This can lead to delays in model deployment and responsiveness to evolving threats. Future work may explore more efficient algorithms, distributed computing, or hardware acceleration techniques to mitigate these issues.

## 8 Data Source

Dataset used: https://staff.itee.uq.edu.au/marius/NIDS_datasets/.

## 9 Conclusion

The proposed deep reinforcement learning-based network intrusion classification framework demonstrates a significant advancement over traditional static models by incorporating a Echo State Network. The dynamic adaptability of DRL enables real-time detection of novel and evolving threats, which is critical in today's rapidly changing network environments. By effectively addressing the class imbalance issues inherent in network intrusion datasets through the innovative use of K-means-based data balancing, author has achieved superior performance across various evaluation metrics. The multi-dataset validation confirms the robustness of our approach, highlighting its potential as a reliable solution for

enhancing the security of modern network infrastructures. Future work may explore further optimizations and the integration of additional machine learning techniques to further enhance detection capabilities.

## Author contributions

**Conceptualization:** Dewan Md. Farid.

**Data curation:** Khorshed Alam.

**Formal analysis:** Khorshed Alam.

**Methodology:** Khorshed Alam, Mahbubul Haq Bhuiyan.

**Supervision:** Dewan Md. Farid.

**Validation:** Khorshed Alam.

**Writing – original draft:** Khorshed Alam, Mahbubul Haq Bhuiyan.

**Writing – review & editing:** Dewan Md. Farid.

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
