## [Decision Letter · Decision Letter 0]

8 Jan 2026

Dear Dr. Farid,

Thank you for submitting your manuscript to PLOS ONE. After careful consideration, we feel that it has merit but does not fully meet PLOS ONE’s publication criteria as it currently stands. Therefore, we invite you to submit a revised version of the manuscript that addresses the points raised during the review process.

We look forward to receiving your revised manuscript.

Kind regards,

Kamran Siddique

Academic Editor

PLOS One

Journal Requirements:

4. We note that Figure 1 in your submission contain copyrighted images. All PLOS content is published under the Creative Commons Attribution License (CC BY 4.0), which means that the manuscript, images, and Supporting Information files will be freely available online, and any third party is permitted to access, download, copy, distribute, and use these materials in any way, even commercially, with proper attribution. For more information, see our copyright guidelines: http://journals.plos.org/plosone/s/licenses-and-copyright.

1) You may seek permission from the original copyright holder of Figure 1 to publish the content specifically under the CC BY 4.0 license.

2) If you are unable to obtain permission from the original copyright holder to publish these figures under the CC BY 4.0 license or if the copyright holder’s requirements are incompatible with the CC BY 4.0 license, please either i) remove the figure or ii) supply a replacement figure that complies with the CC BY 4.0 license. Please check copyright information on all replacement figures and update the figure caption with source information.

If applicable, please specify in the figure caption text when a figure is similar but not identical to the original image and is therefore for illustrative purposes only.

5. We notice that your supplementary information is included in the manuscript file. Please remove them and upload them with the file type 'Supporting Information'. Please ensure that each Supporting Information file has a legend listed in the manuscript after the references list.

**Additional Editor Comments:**

Based on the reviewers’ comments, the manuscript cannot be accepted in its current form. However, I invite the authors to submit a major revision that carefully and comprehensively addresses all reviewer concerns.

Reviewers' comments:

Reviewer's Responses to Questions

**Comments to the Author**

1. Is the manuscript technically sound, and do the data support the conclusions?

Reviewer #1: Yes

Reviewer #2: Partly

2. Has the statistical analysis been performed appropriately and rigorously?

Reviewer #1: Yes

Reviewer #2: Yes

3. Have the authors made all data underlying the findings in their manuscript fully available?

Reviewer #1: Yes

Reviewer #2: Yes

4. Is the manuscript presented in an intelligible fashion and written in standard English?

Reviewer #1: Yes

Reviewer #2: No

Reviewer #1: Major Concerns

1. Novelty and Contribution (Critical)

ESN integration lacks theoretical justification over LSTM/GRU

Limited explanation of why ESN is "highly underexplored" in NIDS

Contribution claims overlap with existing DRL-based IDS work [14, 15, 20]

2. Experimental Design (Critical)

Zero-day attack evaluation (Table 3) is unclear—how does excluding DoS from training prove adaptive modeling when other DoS variants exist?

No computational complexity analysis despite claiming ESN is "lightweight"

Missing real-time latency measurements

Statistical significance tests absent for performance comparisons

3. Methodological Issues (Major)

Reward mechanism (+1/-1) overly simplistic—no justification

Hyperparameter selection (γ=0.96, ϵ=0.07) not explained

Table 7: K-means SMOTE comparison lacks confidence intervals

DRL training convergence criteria vague

4. Presentation Quality (Major)

Excessive background material (Section 2, 3)

Grammatical errors throughout

Algorithm pseudocode formatting inconsistent

Repetitive statements about DRL advantages

Minor Issues

Table 9: Unfair comparison (different training protocols unclear)

LIME analysis (Fig 2) superficial—lacks actionable insights

Missing ablation studies on ESN layer contribution

Related work section too lengthy

Reviewer #2: 1) Based on the results reported in Table 1, the proposed method does not demonstrate a clear improvement over the approaches presented in [15] and [19]. The authors are encouraged to more clearly highlight the novelty and advantages of their method compared to these existing works.

2) The manuscript appears to be excessively long relative to the level of technical novelty it offers. A substantial portion of the paper is devoted to revisiting fundamental concepts of Deep Reinforcement Learning (DRL), which would be more appropriate for a technical report or tutorial-style paper rather than a research-oriented scientific article. The authors are advised to significantly streamline the background section and focus more on the original contributions.

3) One of the major challenges in DRL is the trade-off between exploration and exploitation. The current manuscript does not adequately address how this challenge is handled within the proposed method. A clearer explanation or discussion of this aspect is necessary to strengthen the methodological contribution.

4) To improve the reproducibility of the proposed approach, the authors are strongly encouraged to include a detailed description of the DRL network architecture employed in the study, such as the network structure, key hyperparameters, and training configuration.

5) The authors should ensure that all abbreviations and acronyms are clearly defined upon their first occurrence in the manuscript. Several abbreviations, including NID, AI, and DRL, are used without prior explanation, which may hinder readability for a broader audience.

.

Reviewer #1: **Yes:** Md Mehedi HasanMd Mehedi HasanMd Mehedi HasanMd Mehedi Hasan

Reviewer #2: No

---

## [Author Response · Author response to Decision Letter 1]

23 Feb 2026

Editor’s Comments:

1. Please ensure that your manuscript meets PLOS ONE's style requirements, including those

for file naming. The PLOS ONE style templates can be found at

and

https://journals.plos.org/plosone/s/file?id=ba62/PLOSOne_formatting_sample_title_authors_affili

ations.pdf

2. Please update your submission to use the PLOS LaTeX template. The template and more

information on our requirements for LaTeX submissions can be found at

http://journals.plos.org/plosone/s/latex.

Author’s response: We have used the template based on the links you provided.

3. When completing the data availability statement of the submission form, you indicated that

you will make your data available on acceptance. We strongly recommend all authors decide on

a data sharing plan before acceptance, as the process can be lengthy and hold up publication

timelines. Please note that, though access restrictions are acceptable now, your entire data will

need to be made freely accessible if your manuscript is accepted for publication. This policy

applies to all data except where public deposition would breach compliance with the protocol

approved by your research ethics board. If you are unable to adhere to our open data policy,

please kindly revise your statement to explain your reasoning and we will seek the editor's input

on an exemption. Please be assured that, once you have provided your new statement, the

assessment of your exemption will not hold up the peer review process.

Author’s response: We have added the source of the dataset in a separate new section

named “Data Source”. Previously, we have added the references of the dataset, which is why

it was kept unchanged.

4. We note that Figure 1 in your submission contain copyrighted images. All PLOS content is

published under the Creative Commons Attribution License (CC BY 4.0), which means that the

manuscript, images, and Supporting Information files will be freely available online, and any

third party is permitted to access, download, copy, distribute, and use these materials in any

way, even commercially, with proper attribution. For more information, see our copyright

guidelines: http://journals.plos.org/plosone/s/licenses-and-copyright. We require you to either (1)

present written permission from the copyright holder to publish these figures specifically under

the CC BY 4.0 license, or (2) remove the figures from your submission.

Author’s response: We have removed the figure 1 as due to submitting revised manuscript

time constraints, we are unable to reach the Flaticon. However, our methodology is written in

such a way readers will not require a high level methodology figure.

Reviewer #1: Major Concerns

1. Novelty and Contribution (Critical)

ESN integration lacks theoretical justification over LSTM/GRU

Contribution claims overlap with existing DRL-based IDS work [14, 15, 20]

Author’s response: We included theoretical justification of using ESN in Section 4.3.3.

Regarding the contribution claim, the author acknowledges DRL based IDS work in [14,15,20]

contains similar contributions. However, we proposed a lightweight DRL agent which can predict

unseen attacks shown in section 5. Yes, benchmark models like LSTM can be more accurate

than proposed ESN, but LSTM does require huge computational overhead. Our goal was to

reach similar LSTM level performance by considering less computational units.

2. Zero-day attack evaluation (Table 3) is unclear—how does excluding DoS from training prove

adaptive modeling when other DoS variants exist?

No computational complexity analysis despite claiming ESN is "lightweight"

Missing real-time latency measurements.

Author response: Our evaluation in Table 3 is a targeted simulation of real-world zero-day

conditions, designed to test the model's ability to recognize novel attack variants, not merely

unfamiliar categories. By training on a subset of DoS behaviors (e.g., DDoS) and testing on

withheld DoS subtypes (e.g., GoldenEye), we force the model to learn the fundamental,

transferable features of an attack class rather than overfitting to specific signatures. This

intra-category generalization is a stricter and more meaningful test for a zero-day classifier: if a

model cannot recognize a new variant of a known threat type (e.g., a novel DoS technique), it

will certainly fail against entirely novel threat categories. Therefore, this split strategy directly

evaluates the adaptive, feature-based reasoning essential for detecting true zero-day attacks in

evolving network environments.

Thank you for the valuable feedback regarding computational complexity. You are correct that a

thorough analysis strengthens the claim of being "lightweight." While real-time latency can vary

significantly based on hardware, software stack, and system load making cross-study

comparisons difficult, we have provided a more standardized and hardware-agnostic metric:

FLOPs (Floating Point Operations) in table 10.

3. Reward mechanism (+1/-1) overly simplistic—no justification

Author response: The core contribution of our work is the novel application of an RL-driven

model adaptation framework to the zero-day IDS problem. The simple reward function allows us

to isolate and demonstrate the efficacy of this framework itself, without conflating results with the

effects of a highly tuned reward signal. The strong performance achieved (as shown in our

results) validates that this simple signal is sufficient for the agent to learn an effective policy for

our task.

4. DRL training convergence criteria vague.

Author's response: Thank you for your insightful comment regarding the need for explicit

convergence criteria. In response to your feedback, we have added a dedicated subsection in

the methodology that details the empirical basis for our training duration and convergence

assessment. Our DRL agent was trained for a fixed number of episodes (EPISODES = 31), a

value determined through careful observation of training dynamics, where both reward signals

and validation accuracy consistently plateaued after approximately 15–20 episodes with

negligible subsequent improvement. This conservative episode limit ensures stable

convergence while highlighting the sample efficiency of our ESN-enhanced architecture, which

achieves robust policy optimization rapidly. We believe this clarification strengthens the

reproducibility and methodological transparency of our work.

Reviewer #2:

1. Based on the results reported in Table 1, the proposed method does not demonstrate a clear

improvement over the approaches presented in [15] and [19]. The authors are encouraged to

more clearly highlight the novelty and advantages of their method compared to these existing

works.

Author response: Regarding the contribution claim, the author acknowledges DRL based IDS

work in [15,19] contains similar contributions. However, we proposed a lightweight DRL agent

which can predict unseen attacks shown in section 5. Yes, benchmark models like LSTM can be

more accurate than proposed ESN, but LSTM does require huge computational overhead. Our

goal was to reach similar LSTM level performance by considering less computational units. We

have added FLOPs score table and provided an explanation to how our proposed model is

better in terms of accuracy-computational efficiency trade off.

2. The manuscript appears to be excessively long relative to the level of technical novelty it

offers. A substantial portion of the paper is devoted to revisiting fundamental concepts of Deep

Reinforcement Learning (DRL), which would be more appropriate for a technical report or

tutorial-style paper rather than a research-oriented scientific article. The authors are advised to

significantly streamline the background section and focus more on the original contributions.

Author response: Our writing style was intentionally tailored to align with the multidisciplinary

and accessibility-focused ethos of PLOS ONE, which explicitly encourages manuscripts to be

understandable to researchers across fields. Given that our work sits at the intersection of

network security, machine learning, and reinforcement learning, we included a thorough yet concise background on DRL to ensure that readers from any of these disciplines can fully engage with our contributions without needing to consult external tutorials. That said, we

recognize the importance of conciseness in research articles. We made changes according to

the reviewer's feedback. We believe the adjustments preserve the manuscript’s clarity for a

broad audience.

3. To improve the reproducibility of the proposed approach, the authors are strongly encouraged

to include a detailed description of the DRL network architecture employed in the study, such as

the network structure, key hyperparameters, and training configuration.

Author response: Thank you for emphasizing the importance of reproducibility. We appreciate

your feedback and would like to clarify that the detailed description of the DRL network

architecture, including the network structure, key hyperparameters, and training configuration,

was already provided in Sections 4 of the submitted manuscript.

---

## [Decision Letter · Decision Letter 1]

13 Mar 2026

A Deep Reinforcement Based Echo State Network for Network Intrusion Classification

PONE-D-25-48877R1

Dear Dr. Farid,

We’re pleased to inform you that your manuscript has been judged scientifically suitable for publication and will be formally accepted for publication once it meets all outstanding technical requirements.

Kind regards,

Kamran Siddique

Academic Editor

PLOS One

Additional Editor Comments (optional):

Reviewers' comments:

Reviewer's Responses to Questions

**Comments to the Author**

Reviewer #1: All comments have been addressed

Reviewer #2: All comments have been addressed

2. Is the manuscript technically sound, and do the data support the conclusions?

Reviewer #1: Yes

Reviewer #2: Yes

3. Has the statistical analysis been performed appropriately and rigorously?

Reviewer #1: Yes

Reviewer #2: Yes

4. Have the authors made all data underlying the findings in their manuscript fully available?

Reviewer #1: Yes

Reviewer #2: Yes

5. Is the manuscript presented in an intelligible fashion and written in standard English?

Reviewer #1: Yes

Reviewer #2: Yes

Reviewer #1: author address all of the major concern. i am pleased to accept this manuscript to publish. Before submitting final. author needs to check grammatical errors.

Reviewer #2: The authors addressed all my comments.

I don't have any further comments on the paper.

.

Reviewer #1: No

Reviewer #2: No

---

## [Editor Report · Acceptance letter]

PONE-D-25-48877R1

PLOS One

Dear Dr. Farid,

I'm pleased to inform you that your manuscript has been deemed suitable for publication in PLOS One. Congratulations! Your manuscript is now being handed over to our production team.

Kind regards,

on behalf of

Dr. Kamran Siddique

Academic Editor

PLOS One